# Identification of a missense variant in *SPDL1* associated with idiopathic pulmonary fibrosis

Ryan S. Dhindsa [1], Johan Mattsson[2], Abhishek Nag[1], Quanli Wang[1], Louise V. Wain [3,4], Richard Allen [3], Eleanor M. Wigmore[1], Kristina Ibanez[1], Dimitrios Vitsios[1], Sri V. V. Deevi [1], Sebastian Wasilewski[1], Maria Karlsson[5], Glenda Lassi[6], Henric Olsson[2], Daniel Muthas[2], Susan Monkley[2], Alex Mackay[2], Lynne Murray[5], Simon Young[7], Carolina Haefliger [1], FinnGen Consortium*, Toby M. Maher [8,9], Maria G. Belvisi [10,11,12], Gisli Jenkins [13,14], Philip L. Molyneaux [8,10,15✉], Adam Platt [6,15✉] & Slavé Petrovski[1,15✉]

Idiopathic pulmonary fibrosis (IPF) is a fatal disorder characterised by progressive, destructive lung scarring. Despite substantial progress, the genetic determinants of this disease remain incompletely defined. Using whole genome and whole exome sequencing data from 752 individuals with sporadic IPF and 119,055 UK Biobank controls, we performed a variant-level exome-wide association study (ExWAS) and gene-level collapsing analyses. Our variant-level analysis revealed a novel association between a rare missense variant in *SPDL1* and IPF (NM_017785.5:g.169588475 G > A p.Arg20Gln; $p = 2.4 \times 10^{-7}$, odds ratio = 2.87, 95% confidence interval: 2.03–4.07). This signal was independently replicated in the FinnGen cohort, which contains 1028 cases and 196,986 controls (combined $p = 2.2 \times 10^{-20}$), firmly associating this variant as an IPF risk allele. *SPDL1* encodes Spindly, a protein involved in mitotic checkpoint signalling during cell division that has not been previously described in fibrosis. To the best of our knowledge, these results highlight a novel mechanism underlying IPF, providing the potential for new therapeutic discoveries in a disease of great unmet need.

[1] Centre for Genomics Research, Discovery Sciences, BioPharmaceuticals R&D, AstraZeneca, Cambridge, UK. [2] Translational Science & Experimental Medicine, Research and Early Development, Respiratory and Immunology, BioPharmaceuticals R&D, AstraZeneca, Gothenburg, Sweden. [3] Genetic Epidemiology Group, Department of Health Sciences George Davies Centre, University of Leicester, Leicester, UK. [4] National Institute for Health Research, Leicester Respiratory Biomedical Research Centre, Glenfield Hospital, Leicester, UK. [5] Lung Regeneration, Research and Early Development, Respiratory and Immunology, BioPharmaceuticals R&D, AstraZeneca, Cambridge, UK. [6] Translational Science & Experimental Medicine, Research and Early Development, Respiratory and Immunology, BioPharmaceuticals R&D, AstraZeneca, Cambridge, UK. [7] Precision Medicine and Biosamples, Oncology R&D, AstraZeneca, Cambridge, UK. [8] Royal Brompton Hospital, London, UK. [9] Hastings Centre for Pulmonary Research and Division of Pulmonary, Critical Care and Sleep Medicine, Keck School of Medicine, University of Southern California, Los Angeles, CA, USA. [10] National Heart and Lung Institute, Imperial College, London, UK. [11] Research and Early Development, Respiratory and Immunology, BioPharmaceuticals R&D, AstraZeneca, Gothenburg, Sweden. [12] Respiratory Pharmacology Group, London, UK. [13] Respiratory Research Unit, Division of Respiratory Medicine, University of Nottingham, Nottingham, UK. [14] National Institute for Health Research, Nottingham Biomedical Research Centre, Nottingham University Hospitals NHS Trust, Nottingham, UK. [15] These authors contributed equally: Philip L. Molyneaux, Adam Platt, and Slavé Petrovski. *A list of members and their affiliations appears in the Supplementary Information. ✉email: p.molyneaux@imperial.ac.uk; Adam.Platt@astrazeneca.com; slav.petrovski@astrazeneca.com

diopathic pulmonary fibrosis (IPF) is a progressive scarring disorder of the lung that preferentially affects individuals over the age of 70[1]. Though the mechanisms underlying IPF are unclear, the disease is believed to result from repetitive micro-injuries to the alveolar epithelium that trigger aberrant wound-healing responses. This leads to excessive formation of dense fibrotic tissue that reduces lung compliance and inhibits gas transfer. Approved drugs are not curative and are poorly tolerated due to being associated with considerable side effects[2]. In the absence of a lung transplant, individuals with IPF have an average life expectancy of three to five years after diagnosis[3]. Identifying genetic risk factors of IPF provides valuable insight into disease aetiology, which is a crucial step in the development of more precise therapies. Furthermore, an improved understanding of genetic risk factors associated with IPF may enable stratification of patients in clinical trials[2].

Genome-wide association studies (GWAS) of individuals with sporadic IPF have implicated common variants at several loci containing genes related to lung defence, telomere maintenance, cell-cell adhesion, mTOR signalling, and mitotic spindle assembly[4]. The strongest common variant signal arises from the promoter region of *MUC5B* and confers a roughly five-fold increase in disease risk[5]. Nonetheless, common variants seem to explain a small proportion of IPF heritability compared to rare deleterious variants in the protein-coding region of the genome[6]. Sequencing-based case-control studies have consistently identified three definitive IPF risk genes in both familial and sporadic forms of IPF: *TERT*, *RTEL1*, and *PARN*, all involved in telomerase biology[6,7]. Rare variants in the telomerase RNA component, *TERC*, have also been implicated in sporadic IPF[7]. Despite the remarkable progress in identifying both rare and common variant signals, the underlying genetic predisposition remains unknown for the majority of IPF patients.

In this study, we aimed to identify novel genetic associations with sporadic IPF to improve our understanding of the genetic and molecular architecture of this disease. We conducted the largest exome-based case-control analysis to date by using next generation sequencing data from 752 individuals with sporadic IPF and 119,055 UK Biobank controls screened for non-respiratory disease. This large sample size allowed us to test for both variant- and gene-level associations across the allele frequency spectrum. Some of the results of these studies have been previously reported in the form of an abstract[8].

## Results

**Overview.** We performed variant- and gene-level analyses to identify novel IPF risk factors using sequencing data from 752 European cases with IPF and 119,055 European controls (Fig. 1 and Supplementary Fig. 1a, b). These 752 cases specifically comprised 507 individuals enroled in PROFILE (Prospective Study of Fibrosis In the Lung Endpoints) and 245 UK Biobank participants with IPF (ICD10 code J84*) registered as the primary (Field 40001) or secondary (Field 40002) cause of death (Fig. 1). The median age of diagnosis for the cases was 71 years of age with a median survival of 39.4 months. The control cohort consisted of UK Biobank participants screened for non-respiratory disease (Supplementary Table 1). Although there might be some individuals among the controls that will eventually develop IPF, we expect this error rate to be at most 0.02% given the incidence of IPF in the general European population[9]. Whole-genome sequencing was performed on PROFILE participants and whole-exome sequencing was

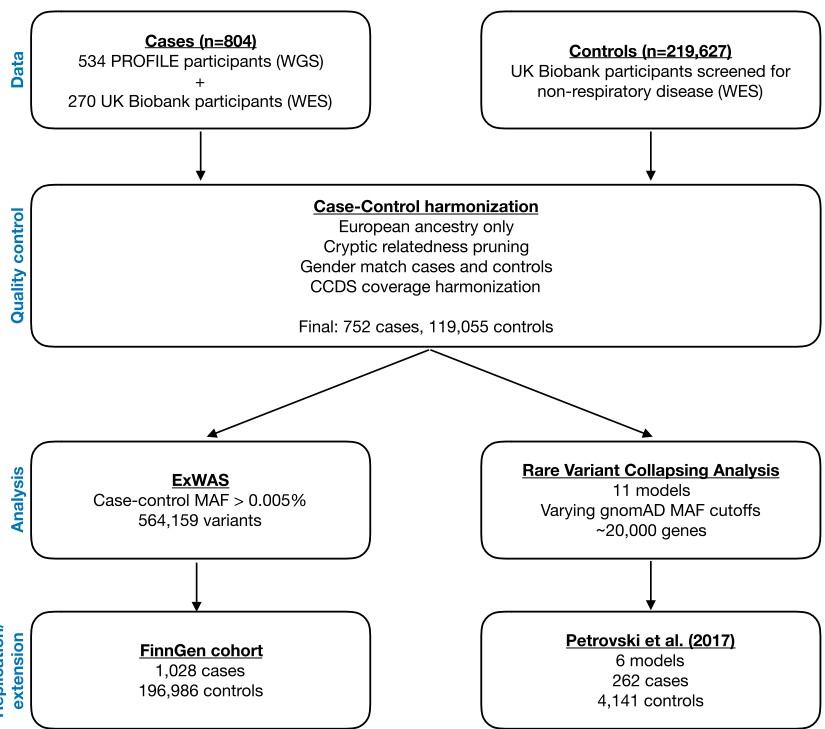

**Fig. 1 Idiopathic pulmonary fibrosis genetic discovery and replication study design.** We combined whole-exome sequence (WES) and whole-genome sequence (WGS) data from a total of 804 cases enroled in either the PROFILE study or the UK Biobank. Controls comprised of 219,627 UK Biobank participants. We harmonised the case-control cohort based on ancestry, relatedness, and gender, resulting in a test cohort total of 752 cases and 119,055 controls screened for non-respiratory disease. We filtered out sites that were differentially covered between cases and controls. We then performed two analyses: an exome-wide variant-level association test (ExWAS) and rare variant collapsing analysis. Variants from the ExWAS with $p < 0.01$ were then reviewed in the FinnGen cohort. Collapsing analyses were also combined with previously published results from an independent case-control study[6]. MAF = minor allele frequency.

performed on UK Biobank participants. Association analyses were limited to pruned protein-coding sequence sites with minimum variability in coverage between cases and controls (Supplementary Table 2). Variant-level associations that achieved a p-value less than 0.01 were reviewed in the FinnGen cohort, which includes genotype data for individuals of Finnish descent, including 1028 individuals with IPF and 196,986 controls (FinnGen release 5) (Fig. 1). We also performed a combined gene-level collapsing analysis using data from a previously published whole-exome sequencing study that included 262 cases and 4141 controls[6] (Fig. 1).

**Exome-wide variant-level association study (ExWAS).** In an exome-wide variant-level association study (ExWAS), 564,159 protein-coding variants were assessed for association with IPF risk (Fig. 2a, b; Supplementary Data 1). We identified five variants that were genome-wide significant ($p < 5 \times 10^{-8}$), and all were in the vicinity of the well-established *MUC5B* risk allele (rs35705950) (Fig. 2a). The next strongest independent signal emerged from a missense variant in the gene *SPDL1* (NM_017785.5:g.169588475 G > A, p.Arg20Gln [rs116483731]; Fisher's exact test [FET] $p = 2.4 \times 10^{-7}$), with an allele frequency of 2.2% in cases compared to 0.78% in controls (odds ratio [OR], 2.87; 95% confidence interval [CI]: 2.03–4.07) (Fig. 2a). This same variant reached genome-wide significance in the FinnGen replication cohort (Supplementary Fig. 2), with a case frequency of 6.9% and a control frequency of 3.0% (logistic regression $p = 1.0 \times 10^{-15}$; OR, 3.13; 95% CI: 2.37–4.14). Taking the combined evidence from both cohorts, this variant achieves a Stouffer Z-test p-value of $2.2 \times 10^{-20}$, unequivocally associating it with IPF risk.

Despite its relatively strong effect size, the *SPDL1* locus has not been previously reported in IPF through prior GWAS with larger sample sizes (Supplementary Table 3)[4]. Because of the contribution of Mendelian genetics to the genetic architecture of IPF, we tested whether the *SPDL1* missense variant may have independently arisen multiple times in Europeans or whether it resides on a common haplotype. In examing haplotypes within the 10 kb window of the *SPDL1* index variant, we found that all *SPDL1* rs116483731 risk allele observations among the PROFILE cohort occurred on a single common ancestral haplotype, which accounted for 16.5% of all haplotypes identified among the 1014 PROFILE chromosomes (Supplementary Fig. 3). This indicates a common ancestral origin for this variant.

**Gene-based collapsing analyses.** Next, we performed gene-based collapsing analyses to identify genes carrying an aggregated excess of rare deleterious variants among the case samples. Despite this being the largest gene-based collapsing test performed in IPF to date, no

new genes reached study-wide significance ($p < 2.4 \times 10^{-7}$) across 11 different rare-variant genetic architectures (Supplementary Fig. 4, Supplementary Table 4 and Supplementary Data 2). We combined these results with data reported in a previously published study of 262 IPF cases and 4141 controls[6], which also did not yield novel study-wide significant findings (Supplementary Fig. 5 and Supplementary Data 3). Next, we explored whether the top-ranked genes that did not meet study-wide significance were enriched for novel, putative disease-associated genes. Mantis-ml[10] identified that case-enriched genes ($p < 0.05$) in the collapsing models that focused on variants in regions intolerant to missense variation[11] were significantly enriched for genes predicted to be associated with pulmonary fibrosis (Supplementary Fig. 6). This result suggests that there are additional IPF risk genes to be discovered in larger case sample sizes.

The primary collapsing model focused on rare (MAF < 0.1%) protein-truncating variants (PTVs). Under this model, the top three genes were the previously reported *RTEL1* (FET $p = 3.0 \times 10^{-7}$; OR, 13.6; 95% CI: 6.6–28.1), *PARN* (FET $p = 2.1 \times 10^{-5}$; OR, 28.9; 95% CI: 9.9–84.2), and *TERT* (FET $p = 8.5 \times 10^{-5}$; OR, 43.3; 95% CI: 12.1–155.7) signals. Given the rarity of this extreme class of variants among these genes in the general population, the effect size of carrying a PTV in these genes conferred larger effect sizes than the more common variants implicated in IPF (Fig. 3a). Notably, the *MUC5B* and *SPDL1* SNPs represented the next strongest risk factors (Fig. 3a).

**Genetic architecture of IPF.** It is well-established that the *MUC5B* promoter risk allele frequency is significantly enriched in cases carrying rare variants in *RTEL1*, *TERT*, and *PARN* compared to controls, albeit at a lower rate than among noncarrier cases[6,12]. Using the WGS data available for the cases in the PROFILE cohort, the *MUC5B* allele frequency in carriers of rare variants in *RTEL1, PARN, TERT*, and *TERC* (21%) was found to be significantly higher than the allele frequency in non-Finnish European controls (11%; FET $p = 0.02$; OR = 2.13; 95% CI: 1.06–3.98), but lower than IPF cases without an identified rare genetic risk factor in a telomere-related gene (33%; FET $p = 0.02$; OR = 0.48; 95% CI: 0.23–0.91) (Fig. 3b and Supplementary Data 4). The *MUC5B* promoter risk allele frequency was also higher among *SPDL1* risk allele carriers (FET $p = 0.001$; OR = 2.95; 95% CI: 1.48–5.58) as well as all other IPF cases (FET $p = 5.9 \times 10^{-83}$; OR = 4.47; 95% CI: 3.88–5.14) compared to non-Finnish European controls (Fig. 3b and Supplementary Data 4). As previously suggested[6], these results underscore the contribution of an oligogenic architecture contributing to IPF risk.

Both the presence of mutations in telomerase genes such as *TERT, TERC, RTEL1* and *PARN* and quantifiably shorter telomere lengths have been associated with poorer prognosis in individuals

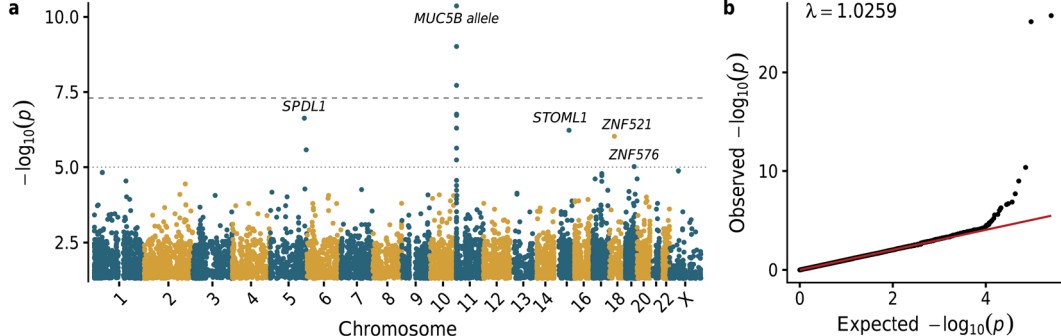

**Fig. 2 Association of single-nucleotide variants with IPF. a** Manhattan plot depicting p-values of the 564,159 exonic variants tested for association with IPF status in 752 cases and 119,055 controls. The long-dash line indicates the genome-wide significance threshold ($p < 5 \times 10^{-8}$). Relevant to the *MUC5B* locus, the Y-axis is capped at $1 \times 10^{-10}$. **b** Quantile–quantile plot of observed versus expected p-values. Linear regression line is indicated in red.

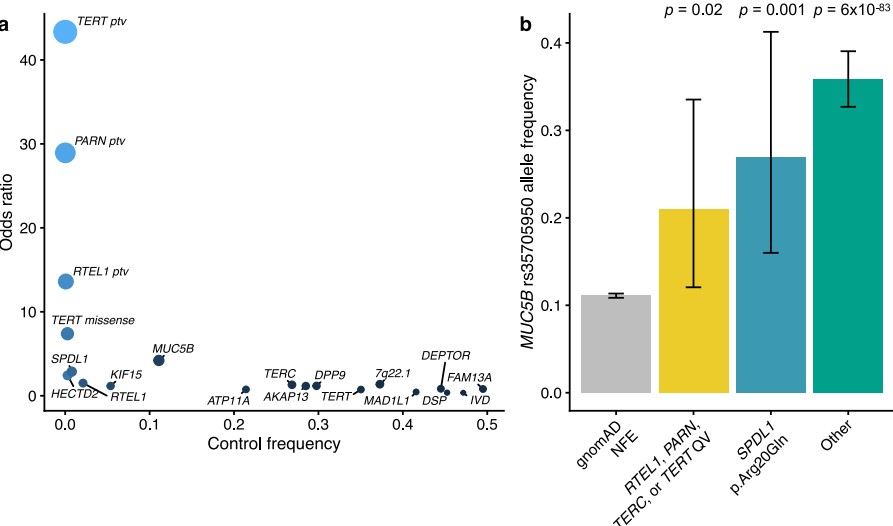

**Fig. 3 Loci associated with IPF. a** Scatter plot depicting the odds ratios versus control frequencies for: protein-truncating variants (PTVs) in *TERT*, *PARN*, and *RTEL1*; rare damaging missense variants in *TERT* based on PROFILE collapsing analyses; the novel missense variant in *SPDL1*; and the sentinel SNPs from the largest IPF GWAS to date[4]. Control frequencies for *TERT*, *PARN*, and *SPDL1* missense reflect the carrier frequencies in the UK Biobank controls used for our association studies. Control frequencies for the remaining alleles were derived from gnomAD[21] non-Finnish European allele frequencies. **b** Allele frequency of the *MUC5B* promoter allele in 32,267 non-Finnish European gnomAD samples and 507 PROFILE IPF cases stratified by genotype. "Other" refers to IPF cases who do not carry a rare variant in *RTEL1*, *PARN*, *TERC*, *TERT*, and do not carry the *SPDL1* missense variant (i.e., noncarriers). Fisher's exact p-values are depicted for comparisons between gnomAD non-Finnish Europeans and QV carriers for each group. Error bars represent 95% confidence intervals.

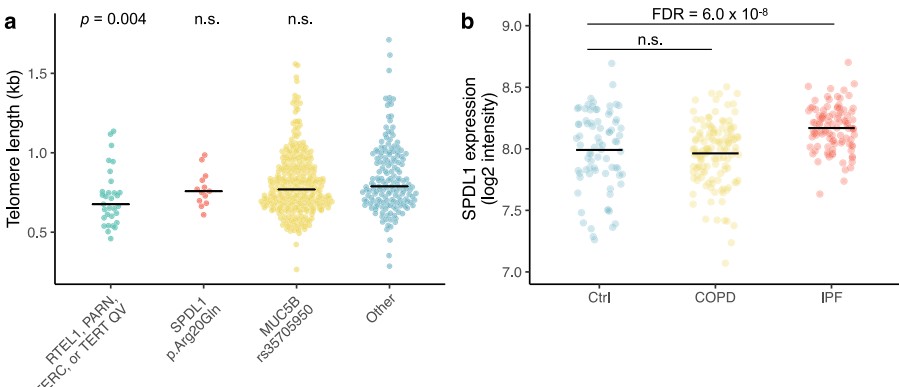

**Fig. 4 Telomere lengths and expression data in individuals with IPF. a** TelSeq-inferred[14] telomere lengths for the 507 IPF cases included in the PROFILE cohort stratified by genetic risk factors. Logistic regression p-values are depicted for each group. QV = qualifying variant, n.s. = non-significant. **b** Expression of *SPDL1* in lung tissue derived from 91 controls, 145 individuals with COPD, and 122 individuals with IPF.

with IPF[13]. Given this, we used TelSeq[14] to infer telomere lengths from the WGS data for the 507 cases included in the PROFILE cohort (Supplementary Data 5). We then tested whether different genetic risk factors were associated with telomere length using a logistic-regression model that included sex and age as covariates. Indeed, the telomeres of PROFILE IPF cases carrying rare putatively pathogenic variants in *RTEL1*, *PARN*, *TERC* or *TERT* were 10–15% shorter than the telomeres of the remaining cases in the PROFILE cohort (logistic regression $p = 0.004$; OR = 0.031; 95% CI: 0.003–0.29) (Fig. 4a; Supplementary Tables 5 and 6). In keeping with prior reports, individuals with these variants were gender balanced (in contrast to the wider PROFILE cohort, which was three quarters male), younger, and had worse survival rates than that seen for the cohort as a whole (Supplementary Table 5 and Supplementary Fig. 7). On the other hand, individuals carrying the minor allele for *MUC5B* or *SPDL1* did not exhibit statistically significant differences in their

telomere lengths compared to the remainder of the PROFILE cohort (Fig. 4a), nor did they exhibit significant differences in demographic or clinical characteristics (Supplementary Table 5). There was no significant enrichment of family history among individuals carrying rare variants in the telomerase genes, the *SPDL1* risk allele, or the *MUC5B* risk allele (Supplementary Table 5). Together, these results suggest that *SPDL1* confers increased IPF risk through a mechanism different from the telomerase pathway genes.

**SPDL1 expression in lung tissue derived from individuals with IPF.** To further interrogate the role of *SPDL1* in the aetiology of IPF, we leveraged a publicly available transcriptomic dataset that includes microarray data derived from lung tissue of 122 patients with IPF, 145 patients with chronic obstructive pulmonary disease (COPD), and 91 controls (GSE47460, GPL14550)[15]. We performed differential gene expression analysis and found that *SPDL1* was

significantly upregulated in IPF tissue compared to controls (1.2-fold increase, FDR = $6.0 \times 10^{-8}$; Fig. 4b and Supplementary Data 6). This pattern was not observed in fresh frozen lung tissue derived from patients with COPD. These results further support the role of *SPDL1* in the pathophysiology of IPF.

## Discussion

This study represents the largest NGS-based case-control analysis of sporadic IPF to date. Our exome-wide association study unveiled a missense variant in *SPDL1* as a novel genetic risk factor for IPF. This variant, which is rare among non-Finnish European controls, confers a larger effect size than more common IPF risk alleles, with the exception of the *MUC5B* allele. Critically, we also found that *SPDL1* is significantly upregulated in lung tissue derived from individuals with IPF. While gene-level collapsing analyses did not reveal novel associations, we observed that nominally significant genes from several collapsing models were significantly enriched for genes predicted to be associated with pulmonary fibrosis. This should encourage ongoing next-generation sequencing of larger case populations. Consistent with previous studies, the results of this study suggest that IPF is characterised by modest locus heterogeneity[6]. The identification of the *SPDL1* variant also underscores the advantage of sequencing-based variant-association tests in capturing signals across the allele frequency spectrum. The higher frequency of this variant in the Finnish population also highlights the value of performing genetic analyses in bottleneck populations.

We found that carriers of rare variants in telomerase pathway genes were significantly younger in age of onset than other IPF cases. However, we did not detect statistically significant differences in clinical features of carriers of either the *SPDL1* or *MUC5B* risk allele compared to the remainder of the cohort. Future studies with larger sample sizes may detect potential differentiating clinical features.

The differences in telomere lengths between *SPDL1*, *MUC5B*, and carriers of ultra-rare variants in telomerase pathway genes may be partially explained by gene function. *RTEL1*, *PARN*, and *TERC*, and *TERT* are all directly involved in the telomerase pathway, whereas *SPDL1* encodes spindly, a coiled-coil domain containing protein that has a critical role in mitosis. This protein regulates chromosome alignment as well as microtubule attachment to kinetochores during prometaphase[16–18]. It also regulates the spindle assembly checkpoint (SAC) and enables kinetochore compaction by recruiting the microtubule motor dynein to kinetochores, facilitating the removal of outer kinetochore components and SAC proteins. Following the formation of stable microtubule attachments, these processes allow cells to progress from metaphase into anaphase and to complete mitosis.

Mechanistically, Spindly functions as an adaptor protein, linking the RZZ complex (Rod, Zw10, and Zwilch) with dynein/dynactin. Spindly binds dynein and dynactin via two C-terminal domains termed the CC1 box and the Spindly Motif respectively. Importantly, cells carrying mutant forms of Spindly lacking these domains have been shown to display aberrant spindle morphology and chromosome segregation errors[17,18]. In addition to its role in mitosis, Spindly has also been reported to localise to the leading edge of migrating cells[19]. Cells either knocked down for Spindly or expressing a mutant version that is deficient in dynactin binding showed reduced migration in a wound scratch assay. This mechanism has been suggested to be involved in the spread of colorectal cancer cells[20].

Recently, two other genes also linked to kinetochore function, *MAD1L1* and *KIF15*, were implicated in an IPF GWAS study[4], suggesting that dysfunction of this pathway may underlie a novel non-telomeric mechanism in IPF. Individuals with the *SPDL1*

minor allele, at least superficially, resemble the wider cohort of IPF subjects. Deeper phenotyping of PROFILE and other IPF cohorts may uncover unique features of disease related to impaired kinetochore function. It will also be important to explore whether genetic mechanisms influence response to anti-fibrotic therapy—a finding which would influence future trial design and precision medicine strategies. Notably, only one individual in the case cohort carried both the *SPDL1* minor allele and a qualifying variation in either *RTEL1*, *PARN*, *TERC*, or *TERT*, further suggesting that these are two independent mechanisms of IPF pathogenesis, and that mitotic spindle dysfunction pathway underlies a novel non-telomeric mechanism in IPF.

Although one might speculate of a role for Spindly in cell senescence or fibroblast migration in pulmonary fibrosis, this finding necessitates further experimental follow-up to understand the pathophysiology of this variant. Importantly, the evidence of multiple pathways for this devastating disorder also emphasizes the need for more targeted, patient-tailored treatments.

## Methods

**Study cohort**. The initial sample consisted of 541 PROFILE cases, 272 UK Biobank cases, and 302,081 UK Biobank controls. The control cohort was restricted to include only individuals without a history of respiratory disease (Supplementary Table 1). We filtered the cohort based on quality control metrics (Supplementary Table 2) and we gender matched the control cohort to the case cohort. The final cohort consisted of 752 cases and 119,055 controls of European ancestry. For the PROFILE cohort, written informed consent was obtained from all subjects and the study was approved by the local research ethics committee (reference numbers 10/H0720/12). Replication analyses were performed using the FinnGen biobank (freeze 5), which includes genotype data for 1028 individuals with IPF and 196,986 controls of Finnish descent.

**Ethics statement**. For the PROFILE cohort written informed consent was obtained from all subjects and the study was approved by the local research ethics committee (reference numbers 10/H0720/12). The UK Biobank WES data described in this paper are publicly available to registered researchers through the UKB data-access protocol. Additional information about registration for access to the data are available at http://www.ukbiobank.ac.uk/register-apply/. Data for this study were obtained under Resource Application Number 26041. The protocols for UK Biobank are overseen by The Ethics Advisory Committee, for more information see https://www.ukbiobank.ac.uk/ethics/ and https://www.ukbiobank.ac.uk/wp-content/uploads/2011/05/EGF20082.pdf

Patients and control subjects in FinnGen provided informed consent for biobank research, based on the Finnish Biobank Act. Alternatively, older research cohorts, collected prior to the start of FinnGen (in August 2017), were collected based on study-specific consents and later transferred to the Finnish biobanks after approval by Fimea, the National Supervisory Authority for Welfare and Health. Recruitment protocols followed the biobank protocols approved by Fimea. The Coordinating Ethics Committee of the Hospital District of Helsinki and Uusimaa (HUS) approved the FinnGen study protocol Nr HUS/990/2017.

The FinnGen study is approved by Finnish Institute for Health and Welfare (THL), approval number THL/2031/6.02.00/2017, amendments THL/1101/5.05.00/2017, THL/341/6.02.00/2018, THL/2222/6.02.00/2018, THL/283/6.02.00/2019, THL/1721/5.05.00/2019, Digital and population data service agency VRK43431/2017-3, VRK/6909/2018-3, VRK/4415/2019-3 the Social Insurance Institution (KELA) KELA 58/522/2017, KELA 131/522/2018, KELA 70/522/2019, KELA 98/522/2019, and Statistics Finland TK-53-1041-17.

The Biobank Access Decisions for FinnGen samples and data utilised in FinnGen Data Freeze 5 include: THL Biobank BB2017_55, BB2017_111, BB2018_19, BB_2018_34, BB_2018_67, BB2018_71, BB2019_7, BB2019_8, BB2019_26, Finnish Red Cross Blood Service Biobank 7.12.2017, Helsinki Biobank HUS/359/2017, Auria Biobank AB17–5154, Biobank Borealis of Northern Finland_2017_1013, Biobank of Eastern Finland 1186/2018, Finnish Clinical Biobank Tampere MH0004, Central Finland Biobank 1–2017, and Terveystalo Biobank STB 2018001.

**Sequencing, alignment, and variant calling**. For the PROFILE cohort, genomic DNA from IPF cases was extracted and underwent paired-end 150 bp WGS at Human Longevity Inc using the NovaSeq6000 platform. For IPF cases, >98% of consensus coding sequence release 22 (CCDS) has at least 10x coverage and average coverage of the CCDS achieved 42-fold read-depth. Genomic DNA from UK Biobank controls underwent paired-end 75 bp whole exome sequencing (WES) at Regeneron Pharmaceuticals using the IDT xGen v1 capture kit on the NovaSeq6000 machines. For UK Biobank controls, >95% of CCDS has at least 10× coverage and average CCDS read-depth of 59X. All case and control sequences were processed

through the same bioinformatics pipeline, this included re-processing all the UK Biobank exomes from their unaligned FASTQ state. A custom-built Amazon Web Services (AWS) cloud compute platform running Illumina DRAGEN Bio-IT Platform Germline Pipeline v3.0.7 was adopted to align the reads to the GRCh38 genome reference and perform small variant SNV and indel calling. SNVs and indels were annotated using SnpEFF v4.3 against Ensembl Build 38.92.

**Cohort pruning**. The initial sample consisted of 541 PROFILE cases, 272 UK Biobank cases, and 302,081 UK Biobank controls (Supplementary Table 2). We removed samples where there was a discordance between self-reported and X:Y coverage ratios, as well as samples with >4% contamination according to VerifyBamID. The cohort was screened with KING to ensure that only unrelated (up to third-degree) individuals were included in the test. To reduce variation due to population stratification, we only included individuals with a probability of European Ancestry ≥0.98 based on PEDDY predictions and individuals within four standard deviations of principal components 1–4 (Supplementary Fig. 1). Further, samples were required to have greater than 95% of CCDS (release 22) bases covered with at least 10-fold coverage.

The control cohort was further restricted to include only individuals without a history of respiratory disease (Supplementary Table 1). Using random sampling of the controls, we gender matched the control cohort to the case cohort (75% male). The final cohort consisted of 752 cases and 119,055 controls.

**Exome-wide association study (ExWAS)**. We tested 564,159 protein-coding variants association with IPF status. We specifically included all variants that were present in at least 12 individuals in the case-control cohort and passed the following QC criteria: minimum coverage 10X; percent of alternate reads in heterozygous variants ≥0.3 and ≤0.8; binomial test of alternate allele proportion departure from 50% in heterozygous state $p > 10^{-6}$; genotype quality score (GQ) ≥ 30; Fisher's strand bias score (FS) ≤ 200 (indels) ≤60 (SNVs); mapping quality score (MQ) ≥ 40; quality score (QUAL) ≥ 30; read position rank sum score (RPRS) ≥ −2; mapping quality rank sum score (MQRS) ≥ −8; DRAGEN variant status = PASS; Binomial test of difference in missingness between cases and controls $p < 10^{-6}$; variant did not achieve Hardy-Weinberg Equilibrium Exact $p < 10^{-5}$; variant site is not missing (i.e., <10X coverage) in ≥1% of cases or controls; variant did not fail above QC in ≥0.5% of cases or controls; variant site achieved 10-fold coverage in ≥50% of GnomAD[21] exomes, and if variant was observed in GnomAD the variant calls in GnomAD achieved exome z-score ≥ −0.2 and exome MQ ≥ 30.

$p$-Values were generated via Fisher's exact test. Replication analyses were performed using the FinnGen dataset (release 5), which includes 1028 cases and 196,986 controls. For all variants achieving a $p$-value less than 0.01 in the ExWAS, we computed combined $p$-values via Stouffer's Z-test and defined genome-wide significance as the conventional $p < 5 \times 10^{-8}$. Furthermore, a firth logistic regression, including sex, age and the top four PC's as covariates was also calculated for the SPDL1 association in the internal analysis, ([rs116483731]; Fisher's exact test [FET] $p = 2.4 \times 10^{-7}$ and Firth's logistic regression p = $7.2 \times 10^{-6}$).

To estimate the expected null distribution given the sparsity of data and heavily imbalanced case-control composition we permuted case and control labels ten times and then re-ran the Fisher's Exact test for each of the ~500 K ExWAS variants. We then rank sorted the p-values from each of the permutation runs and took the median across permutations as the expected $p$-value to define a permutation-based null distribution. We then used the estlambda2 function implemented in the R Package QQPerm[22] with default parameters to compute lambda.

**Collapsing analysis**. To perform collapsing analyses, we aggregate variants within each gene that fit a given set of criteria, identified as qualifying variants (QVs)[6]. We performed 10 non-synonymous collapsing analyses, including 9 dominant and one recessive model, plus an additional synonymous variant model as a negative control. In each model, for each gene, the proportion of cases is compared to the proportion of controls carrying one or more qualifying variants in that gene. The exception is the recessive model, where a subject must have two qualifying alleles. The criteria for qualifying variants in each collapsing analysis model are in Supplementary Table 4. $p$-Values were generated via Fisher's exact test. Additionally, we performed a combined analysis using the Cochran–Mantel–Haenszel test considering six of the collapsing models included in a prior IPF study[6].

For all models (Supplementary Table 4) we applied the following QC filters: minimum coverage 10X; annotation in CCDS transcripts (release 22; ~34 Mb); percent alternate reads in homozygous genotypes ≥ 0.8; percent of alternate reads in heterozygous variants ≥0.3 and ≤0.8; binomial test of alternate allele proportion departure from 50% in heterozygous state $p > 10^{-6}$; genotype quality score (GQ) ≥ 30; Fisher's strand bias score (FS) ≤ 200 (indels) ≤60 (SNVs); mapping quality score (MQ) ≥ 40; quality score (QUAL) ≥ 30; read position rank sum score (RPRS) ≥−2; mapping quality rank sum score (MQRS) ≥−8; DRAGEN variant status = PASS; Binomial test of difference in missingness between cases and controls $p < 10^{-6}$; variant is not missing (i.e., <10X coverage) in ≥1% of cases or controls; variant site achieved 10-fold coverage in ≥25% of GnomAD samples, and if variant was observed in GnomAD the variant calls in GnomAD achieved exome z-score ≥−2.0 and exome MQ ≥ 30.

**SPDL1 haplotype analysis**. We constructed haplotypes for the SPDL1 locus in the PROFILE cohort ($N = 507$) and in a random subset of 25,000 unrelated European individuals from the UK Biobank. This was done by phasing a total of 76 variants (MAF > 1%) that were located within a 10 kb window of rs116483731. The genotype phasing was performed using MACH[23], implementing the following parameters:–states 1000 and–rounds 60. Following the genotype phasing, we estimated the number of distinct haplotypes and their corresponding frequencies in each dataset. Next, in each dataset, we identified the SPDL1 risk haplotypes i.e. haplotypes that contained the risk allele 'A' for rs116483731. Then, for each SPDL1 risk haplotype that was identified, we determined the corresponding ancestral haplotype (having the reference allele 'G' in place of the risk allele 'A' for rs116483731) and estimated the percentage of those with that ancestral haplotype background who carried the rs116483731 mutation.

**Putatively pathogenic RTEL1, PARN, TERC, and TERT variants**. To identify PROFILE subjects with putatively pathogenic variants in RTEL1, PARN, TERT, or TERC we adopted the following criteria:

1. Variants were required to fulfil the following QC thresholds:
   a. Percentage of variant allele reads ≥ 0.3
   b. Binomial exact test of the departure from heterozygous expectation of 0.5 for variant allele read ratio $p > 0.001$
   c. GQ ≥ 30
   d. QUAL ≥ 30
   e. Variant affects a CCDS transcript
2. For putative protein-truncating variants (PTVs) in RTEL1, PARN, and TERT the gnomAD minor allele frequency ≤0.05% (gnomAD popmax)
3. For missense variants in TERT and PARN the gnomAD minor allele frequency = 0 (ultra-rare)
4. For TERC noncoding RNA variants they were annotated by ClinVar as Pathogenic or the same TERC nucleotide was recurring affected by multiple variants.

All putatively pathogenic variants are included in Supplementary Data 5.

**Effect size comparisons**. We compared the effect sizes for different classes of variants implicated in IPF: PTVs in TERT, RTEL1, and PARN; putatively damaging missense variants in TERT, and GWAS loci that reached genome-wide significance in the largest IPF GWAS performed to date[4]. We used the UCSC Genome Browser LiftOver tool to convert the reported GRCh37 coordinates to GRCh38 coordinates, requiring 100% of bases to remap. We excluded the MAPT risk allele (rs2077551), as this allele failed the gnomAD random forest filter. For coding variants in TERT, RTEL1, and PARN, we used the case versus control odds ratios calculated in our collapsing analysis. For the GWAS loci, we used the PROFILE WGS data to calculate the frequency in cases, and we used the gnomAD non-Finnish European allele frequencies to derive the frequency in controls.

**Clinical characteristics**. We compared clinical features for individuals in the PROFILE cohort who carried the MUC5B risk allele, the SPDL1 risk allele, or putatively pathogenic variants in RTEL1, TERC, TERT, and PARN (Supplementary Table 5). For each genotype, carriers were compared to all other individuals in the PROFILE cohort (i.e., non-carriers for the given risk allele). We specifically assessed differences in gender, survival months, sample age, height, weight, forced vital capacity (FVC), and diffusing capacity for carbon monoxide (DLCO). We performed median data-imputation to account for missing data. The p-values for comparing gender imbalances were generated via Fisher's exact test, whereas p-values for all other clinical characteristics were generated via the Mann–Whitney U-test. Mortality rates were compared among carriers of rare variants in TERT, TERC, PARN, and RTEL1, carriers of the SPDL1 risk allele, and carriers of the MUC5B risk allele versus non-carriers using the Kaplain–Meier method and p-values were generated using a log-rank test.

**Telomere length comparisons**. For 96% of the samples the read lengths ranged between 148 to 150 bp. These BAM files were put through computational telomere length prediction method Telseq v0.0.2[14] using a repeat number of 10. WGS sequences in this cohort did not use a PCR-free DNA sequencing protocol. Logistic regression models were used to determine whether there were differences in telomere lengths between carriers and non-carriers for the MUC5B risk allele, the SPDL1 risk allele, and TERT, TERC, RTEL1, or PARN putatively pathogenic variants; age and sex were included as additional covariates in the models:

$$logit\big(Pr\big(D_i = carrier\big)\big) = \beta_0 + \beta_1\big(telomere\,length\big) + \beta_2\big(age\big) + \beta_3\big(sex\big) \quad (1)$$

where $D_i = 1$ if individual $i$ is a carrier of the given genetic risk factor.

**Mantis-ml**. Known pulmonary fibrosis-associated genes were automatically extracted from the Human Phenotype Ontology (HPO) by specifying the following term in the input configuration file of mantis-ml:[10] "Disease/Phenotype terms: pulmonary fibrosis". This resulted in the following 38 HPO-defined seed genes:

ABCA3, SFTPA2, AP3B1, CAV1, DPP9, CFTR, CCR6, CCN2, CTLA4, PTPN22, TINF2, DKC1, DSP, FAM13A, DCTN4, RTEL1, HLA-DPA1, HLA-DPB1, HLA-DRB1, HPS1, IRF5, PARN, PRTN3, FAM111B, RCBTB1, STN1, SFTPA1, NOP10, CLCA4, SFTPC, NHP2, STX1A, ATP11A, MUC5B, TERC, TERT, TGFB1 and HPS4.

Automatic feature compilation on mantis-ml was performed by providing the following "Additional associated terms" in the input configuration file: "pulmonary, respirat and lung." Mantis-ml was trained using six different classifiers: Extra Trees, XGBoost, Random Forest, Gradient Boosting, Support Vector Classifier and feed-forward Deep Neural Net. Once the mantis-ml genome-wide probabilities of being an IPF gene were generated, we performed a hypergeometric test to determine whether the top-ranked collapsing analysis genes (i.e., genes achieving a $p < 0.05$ in the collapsing analyses) were significantly enriched for the top 5% of mantis-ml IPF-predicted genes. A statistically significant result from the hypergeometric test suggests that there are disease-ascertained genes among the top hits of the collapsing results. We then tested for this enrichment after excluding known IPF genes from our collapsing results to determine whether the signal was independent of genes already associated with IPF. In parallel, we also performed the hypergeometric enrichment test using the synonymous collapsing model to define our empirical null distribution. Among all six mantis-ml integrated classifiers used for training, Gradient Boosting achieved the highest enrichment signal against the collapsing analysis results from IPF.

**Differential gene expression analysis.** Raw data from the GSE47460 dataset[15] were downloaded from the Gene Expression Omnibus. We only analysed the microarray data derived from samples that were derived from the SurePrint G3 Human GE 8x60K Microarray (GPL14450), as the majority of these samples had more genes included. The text files obtained from GEO were processed using the Limma package[24], which we used to perform background correction with the 'normexp' method offsetting for internal background measurements. Quantile normalisation between arrays with log2 transformation and then probes were filtered based on the PA-matrix. Samples were excluded if there was a gender mismatch between gene expression and metadata. Additional samples were excluded if they failed QC metrics during processing. We used the Limma moderated t-test in Array Studio to generate differentially expressed genes, comparing the IPF or COPD samples to controls controlling for gender and age. To collapse the probes, the probes with the lowest false discovery rate (FDR) for each gene was selected to generate the final data set of differentially expressed genes. For plotting SPDL1 expression, we focused on the probe A_23_P41948, as this probe exhibited higher expression across the control samples. Although, we confirmed that the alternative probe A_33_P3249354 was highly correlated with selected probe A_23_P41948 (Pearson's $r = 0.83$, $p < 2 \times 10^{-16}$).

**Reporting summary.** Further information on research design is available in the Nature Research Reporting Summary linked to this article.

## Data availability

Data supporting the findings of this study were derived from whole blood samples that were either whole-genome or whole-exome sequenced. Summary statistics from the exome-wide association study (ExWAS) and collapsing analyses are available within the article and its supplementary information files. For access to the UK Biobank, please register and apply through the UK Biobank website: https://bbams.ndph.ox.ac.uk/ams/. For access to the FinnGen data, please apply through the FinnGen website: https://www.finngen.fi/fi.

## Code availability

Sequence data were processed through a custom-built Amazon Web Services (AWS) cloud compute platform running Illumina DRAGEN Bio-IT Platform. SNVs and indels were annotated using SnpEFF v4.3 against Ensembl Build 38.92. The software used in this study are referenced in the manuscript and are available below:

TelSeq: https://github.com/zd1/telseq
mantis-ml: https://github.com/astrazeneca-cgr-publications/mantis-ml-release
QQperm: https://cran.r-project.org/web/packages/QQperm/index.html
Finngen association tests: https://finngen.gitbook.io/documentation/methods/phewas/logistic-regression

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

## Acknowledgements

We thank the participants and investigators in the UK biobank study (Resource Application Number 26041), and PROFILE clinical colleagues who have referred IPF cases. We thank the AstraZeneca Centre for Genomics Research Analytics and Informatics team for processing and analysis of sequencing data. L.W. holds a GSK/British Lung Foundation Chair in Respiratory Research. The research was partially supported by the National Institute for Health Research (NIHR) Leicester Biomedical Research Centre; the views expressed are those of the author(s) and not necessarily those of the National Health Service (NHS), the NIHR or the Department of Health. P.L.M. is supported by an Action for Pulmonary Fibrosis Mike Bray fellowship. T.M.M. is supported by a National Institute for Health Research Clinician Scientist Fellowship (NIHR ref: CS-2013-13-017) and is a British Lung Foundation Chair in Respiratory Research (C17-3). Funding for the sequencing in the PROFILE and UK Biobank cohort was provided fully or partially by AstraZeneca. The FinnGen project is funded by two grants from Business Finland (HUS 4685/31/2016 and UH 4386/31/2016) and eleven industry partners (AbbVie Inc, AstraZeneca UK Ltd, Biogen MA Inc, Celgene Corporation, Celgene International II Sàrl, Genentech Inc, Merck Sharp & Dohme Corp, Pfizer Inc., GlaxoSmithKline, Sanofi, Maze Therapeutics Inc., Janssen Biotech Inc). Following biobanks are acknowledged for collecting the FinnGen project samples: Auria Biobank (www.auria.fi/biopankki), THL

Biobank (www.thl.fi/biobank), Helsinki Biobank (www.helsinginbiopankki.fi), Biobank Borealis of Northern Finland (https://www.ppshp.fi/Tutkimus-ja-opetus/Biopankki/Pages/Biobank-Borealis-briefly-in-English.aspx), Finnish Clinical Biobank Tampere (www.tays.fi/en-US/Research_and_development/Finnish_Clinical_Biobank_Tampere), Biobank of Eastern Finland (www.ita-suomenbiopankki.fi/en), Central Finland Biobank (www.ksshp.fi/fi-FI/Potilaalle/Biopankki), Finnish Red Cross Blood Service Biobank (www.veripalvelu.fi/verenluovutus/biopankkitoiminta) and Terveystalo Biobank (www.terveystalo.com/fi/Yritystietoa/Terveystalo-Biopankki/Biopankki/). All Finnish Biobanks are members of BBMRI.fi infrastructure (www.bbmri.fi).

## Author contributions

R.S.D., M.G.B., T.M.M., P.L.M., A.P., and S.P. conceived and designed the study. R.S.D., A.N., Q.W., K.I., and D.V. performed the statistical and computational analyses. Q.W., E.M.W., S.V.V.D., and S.W. performed bioinformatics processing. S.P., supervised the analyses. L.V.W., G.L., S.Y., C.H., T.M.M., G.J., P.L.M., and A.P participated in the acquisition of the data. R.S.D., L.V.W., R.A., M.K., H.O., A.M., L.M., T.M.M., M.G.B., G.J., P.L.M., A.P., and S.P. participated in the data interpretation. R.S.D., J.M., S.M, and D.M. analysed the transcriptomic data. R.S.D., T.M.M., P.L.M., A.P., and S.P. wrote the initial draft of the manuscript. All authors contributed to the revision of the first draft. All authors approved the final version of the manuscript.

## Competing interests

R.S.D., J.M., A.N., Q.W., E.M.W., D.V., S.V.V.D., S.W., M.K., G.L., H.O., D.M., S.M., A.M., C.H., M.G.B., A.P., and S.P. are current employees and/or stockholders of AstraZenca. G.J. reports personal fees and other from Biogen, personal fees from Galapagos, other from Galecto, personal fees and other from GlaxoSmithKline, personal fees from Heptares, personal fees from Boehringer Ingelheim, personal fees from Pliant, personal fees from Roche/InterMune, personal fees from MedImmune, personal fees from PharmAkea, personal fees from Bristol Myers Squibb, personal fees from Chiesi, personal fees from Roche/Promedior, other from RedX, other from NuMedii, other from Nordic Biosciences, personal fees from Veracyte, outside the submitted work; and G.J. is supported by a National Institute of Health Research Professorship (NIHR ref: RP-2017-08-ST2-014) and is a trustee for Action for Pulmonary Fibrosis. P.L.M. via his institution received industry-academic funding from AstraZeneca and has received speaker and consultancy fees from Boehringer Ingelheim and Hoffman-La Roche outside the submitted work. L.V.W. has received grant funding from GSK and Orion outside of the submitted work. T.M.M. has via his institution, received industry-academic funding from Astra Zeneca and GlaxoSmithKline R&D and has received consultancy or speakers fees from Astra Zeneca, Bayer, Blade Therapeutics, Boehringer Ingelheim, Bristol-Myers Squibb, Galapagos, Galecto, GlaxoSmithKline R&D, IQVIA, Pliant, Respivant, Roche and Theravance. The remaining authors declare no competing interests.
