## [Peer Review File · Communications Biology]

Reviewers' comments:

Reviewer #1 (Remarks to the Author):

This paper focus on the genetic pathogenic-associated factors of idiopathic pulmonary fibrosis and identify a novel missense variant in SPDL1 by the application of exome-wide association study (exWAS) and gene-level collapsing analyses. Research object are sporadic IPF patients and controls and get a conclusion that SPDL1 is associated with fibrosis. The design of the research is rigorous and the completion is somehow meaningful in the genetics research of IPF.

However, there are still some problems need to be solved before the publication.

Major:

First, compared to MUC5B risk allele (rs35705950), the frequency of variant SPDL1(NM_017785.5 p.Arg20Gln) seems to be much lower , this result may need more basic clinical studies to enhance the reliability of the variant in SPDL1. To identify SPDL1 associate with IPF, the expression of SPDL1 protein in IPF patients and controls, especially in the lung tissues should be tests and future statistical analysis should be conducted.

Second, to prove the reliability of difference between the IPF patients and "controls", the clinical data of controls should be provided to exclude the recognized related factors of IPF ,such as ages, smoking, occupational risk factors etc.

Third, as the MUC5B promoter risk allele frequency is significantly enriched in cases carrying rare variants in RTEL1, TERT, and PARN compared to controls, while the SPDL1 risk allele carrier do not show a similar trend, do MUC5B promoter risk allele frequency or the SPDL1 risk allele has any association with the other genetic pathogenic factors- Surfactant protein mutations such as ABCA3 、 SFTPA2、 SFTPC etc?

Fourth, as the novel missense variant in SPDL1 is more or less associated with the occurrence of IPF , what's the clinical difference between the variant-carries and non- carries? Discuss the probable factors that result in the clinical difference.

Minors:

The variant SPDL1(NM_017785.5 p.Arg20Gln) should also be described on the genomic or transcript level according to the HGVS nomenclature guidelines.

In the result section of exWAS, it is better to present the results of other statistically significant genes in exWAS study.

Line 65: The description "The five genome-wide-significant variants" should be more detailed, does it mean the five variants in the first signal, or the variants reach the minimal statistical significance?

There are still some grammar defects such as Line 65: "assoessed" should be "assessed".

Reviewer #2 (Remarks to the Author):

Dhindsa et al. conducted a exome-wide association study using WGS and WES data and replicated a novel rare variant (p.Arg20Gln) in SPDL1 for idiopathic pulmonary fibrosis risk. This is an interesting study examining rare variants, though results themselves are not particularly novel (one novel variant). This manuscript is well written. Specific comments noted below for possible revision:

1) Fisher's exact test was the only test used for generating p-values. Although Fisher's exact test is of interest for rare variants and small sample size, it also has notable limitations such as not being able to incorporate covariates such as age, sex and PCs. Another thing the authors might want to take into consider for statistical testing is the extremely unbalanced case-control (752 cases: 119,055 controls) design. I'd recommend some methods that take care of it such as methods that implemented saddle point approximations.

2) It was unclear to me exactly how/why the authors decided to use the criteria as they defined in the paper to select putatively pathogenic variants.

3) Although the authors make a good point that the novel variant in SPDL1 may not involve in the telomerase pathway, by showing that carriers of MUC5B or SPDL1 did not exhibit statistically significant differences in their telomere lengths compared to the remainder of participants. I still feel that stronger, more direct evidence are needed for explaining the possible new etiology for SPDL1.

Reviewer #1 (Remarks to the Author):

This paper focus on the genetic pathogenic-associated factors of idiopathic pulmonary fibrosis and identify a novel missense variant in *SPDL1* by the application of exome-wide association study (exWAS) and gene-level collapsing analyses. Research object are sporadic IPF patients and controls and get a conclusion that *SPDL1* is associated with fibrosis. The design of the research is rigorous and the completion is somehow meaningful in the genetics research of IPF. However, there are still some problems need to be solved before the publication.

We thank the reviewer for their positive feedback on the “rigorous” nature of our analytical approach and the view that this research is “meaningful in the genetics research of IPF.”

Major:

First, compared to *MUC5B* risk allele (rs35705950), the frequency of variant *SPDL1*(NM_017785.5 p.Arg20Gln) seems to be much lower, this result may need more basic clinical studies to enhance the reliability of the variant in *SPDL1*. To identify *SPDL1* associate with IPF, the expression of *SPDL1* protein in IPF patients and controls, especially in the lung tissues should be tests and future statistical analysis should be conducted.

We have now studied the publicly available gene expression dataset (GSE47460), which includes data derived from lung tissue of 254 patients with IPF, 220 patients with COPD, and 108 controls. Critically, we find that *SPDL1* is significantly upregulated in the IPF tissue compared to control tissue (1.2-fold increase, FDR $p = 6 \times 10^{-8}$). This functional data from IPF lung tissue further corroborates the role of *SPDL1* in the pathophysiology of IPF and incorporating this reviewer’s suggestion has strengthened our report. We include this new analysis into the revised manuscript (Fig 4b; pg 8, lines 137-150).

Similar to the well-known *MUC5B* allele, which to this day remains a hot area of mechanistic research, there is a long way to go to fully elucidate the causal disease mechanism of the newly identified *SPDL1* IPF missense variant. We hope that sharing this genetic finding will motivate further experimental work of the *SPDL1* variant (NM_017785.5 p.Arg20Gln) to understand disease mechanism not only by us, but also the wider scientific community.

Second, to prove the reliability of difference between the IPF patients and “controls”, the clinical data of controls should be provided to exclude the recognized related factors of IPF, such as ages, smoking, occupational risk factors etc.

Table S1 outlines how we carefully filtered our controls to exclude UKB participants reporting any respiratory diagnosis: IPF, COPD or other. This strict respiratory-disease exclusion criteria

reduced the available UK Biobank control cohort by ~27% (Table S2). We have now also acknowledged in the revised text that it remains possible that individuals among our adopted controls may eventually develop IPF. However, with the prevalence of IPF in the general European population at ~0.02%, this would reflect at most a 0.02% error rate in controls¹. This has been added as a discussion point in the manuscript (pg. 4, Lines 54-56).

Third, as the MUC5B promoter risk allele frequency is significantly enriched in cases carrying rare variants in RTEL1, TERT, and PARN compared to controls, while the SPDL1 risk allele carrier do not show a similar trend, do MUC5B promoter risk allele frequency or the SPDL1 risk allele has any association with the other genetic pathogenic factors- Surfactant protein mutations such as ABCA3, SFTPA2, SFTPC etc?

This important contrast could have been made clearer in our Fig 3B. Here, we compared the MUC5B promoter risk allele enrichment across the three IPF groups (RTEL1, PARN, TERT and TERC mutation carriers [group 1], SPDL1 p.Arg20Gln variant carriers [group 2] and all other IPF cases [group 3]) to the gnomAD non-Finnish European allele frequency. Even among SPDL1 carriers, we find a significant enrichment of the MUC5B promoter risk allele (p=0.001). We now include the Fisher's exact allelic test statistics to Fig 3b and expand in the results (page 7, lines 115-117).

Fourth, as the novel missense variant in SPDL1 is more or less associated with the occurrence of IPF, what's the clinical difference between the variant-carriers and non- carriers? Discuss the probable factors that result in the clinical difference.

The reviewer raises a very interesting question. To address this we considered a variety of available clinical features, including sex, survival, height, weight, forced vital capacity, DLCO, and family history (Table S7). Comparing these characteristics between SPDL1 p.Arg20Gln carriers and non-carriers, none of these comparisons reached significance. Lack of significant clinical differentiating factors was also the case for MUC5B carriers vs non-carriers. However, we did find that carriers of mutations in telomerase dysfunction genes were significantly younger in age of onset and had significantly smaller telomere lengths than the other IPF cases (lines 119-136). Future studies with larger samples will be better positioned to determine if SPDL1 carriers have any clear differentiating clinical factors. We now draw readers to Table S7 and expand on this in the manuscript discussion (pg. 10, lines 195-199).

Minors:

The variant SPDL1(NM_017785.5 p.Arg20Gln) should also be described on the genomic or transcript level according to the HGVS nomenclature guidelines.

This is now introduced in the first mention of the variant (pg 5, line 70).

In the result section of exWAS, it is better to present the results of other statistically significant genes in exWAS study.

The only other statistically significant variants in the exWAS were at the MUC5B locus, this is described on lines 67-69. Genes that fell below the genome-wide significance threshold are annotated in the Manhattan Plot in Fig 2a.

Line 65: The description "The five genome-wide-significant variants" should be more detailed,

does it mean the five variants in the first signal, or the variants reach the minimal statistical significance?

We now clarify that there were a total of five genome-wide significant variants, and all of them fell within the vicinity of the established *MUC5B* risk allele (lines 67-68). When meta-analysing with FinnGen data for all ExWAS variants achieving a $P < 0.01$, only *SPDL1* achieved genome-wide significance (pg 13, lines 284-287).

There are still some grammar defects such as Line 65: “assoessed” should be “assessed”. We have made this correction and looked through the manuscript for any additional events.

Reviewer #2 (Remarks to the Author):

Dhindsa et al. conducted a exome-wide association study using WGS and WES data and replicated a novel rare variant (p.Arg20Gln) in *SPDL1* for idiopathic pulmonary fibrosis risk. This is an interesting study examining rare variants, though results themselves are not particularly novel (one novel variant). This manuscript is well written. Specific comments noted below for possible revision:

1) Fisher’s exact test was the only test used for generating p-values. Although Fisher’s exact test is of interest for rare variants and small sample size, it also has notable limitations such as not being able to incorporate covariates such as age, sex and PCs. Another thing the authors might want to take into consider for statistical testing is the extremely unbalanced case-control (752 cases: 119,055 controls) design. I’d recommend some methods that take care of it such as methods that implemented saddle point approximations.

We thank the reviewer for raising this point. As the reviewer mentions, an exact test is, by design, the most robust statistical approach for studying sparse and extremely unbalanced case-control designs. Our experiences with large-scale rare-variant case-control studies have identified that regression approaches are not a superior substitute for robustness of an exact test^{2,3}.

The genomic inflation factor (λ_{GC}) has proven to be a key guiding metric of whether there is underlying systematic bias in the test statistic distribution. Throughout the collapsing analyses (Fig S4) the highest lambda inflation factor was a satisfactory 1.07. In working through the responses we notice that we unintentionally omitted the QQ plot and corresponding λ_{GC} for our ExWAS findings. This is now included in our revised manuscript. In brief, with >500K variants studied, the genomic inflation factor (λ_{GC}) for our ExWAS is 1.026, Fig 2B. All our lambdas are well in-line with community standards, particularly for rare-variant case-control imbalanced cohort studies² and suggest no underlying systematic bias influencing the test statistic distribution.

The healthy lambda statistics reflect the carefulness adopted in our pre-association statistics where we harmonize the case-control composition in this study for sex, sequencing coverage, and population stratification adopting the procedures outlined in Table S2. For example, our control cohort was, by design, sex-matched (OR = 1.00, Fisher's P = 0.97). We restricted to individuals with a probability of European Ancestry ≥ 0.98 and among those only individuals within four standard deviations of principal components 1-4 (see also Fig. S1). Finally, as we focus on inherited QV's there is no dependency on an individual's age; however, there is the possibility that there are some individuals among our controls who could still develop IPF; however, given the IPF prevalence in the general European population, this would reflect at most a 0.02% control misclassification rate and would also have a conservative effect on test statistics. We have highlighted this potential control misclassification rate in our revised manuscript (pg. 4, Lines 54-56).

2) It was unclear to me exactly how/why the authors decided to use the criteria as they defined in the paper to select putatively pathogenic variants.

We expanded the text describing our criteria for determining putatively pathogenic variants. Firstly, we adopted stringent variant-call QC metrics (outlined in methods). We then adopted conservative population frequency thresholds. We based our thresholds on previous work in an independent US-recruited IPF cohort that illustrated that IPF risk is most highly concentrated at ultra-rare frequencies (See Petrovski et al., 2015; PMID: 28099038). For *TERC*, which is a noncoding RNA, we had less information about functional effects of individual variants so we conservatively only included those that were previously annotated in ClinVar as being "pathogenic" or *TERC* nucleotides that were recurrently mutated among the Imperial-PROFILE IPF patient collection.

3) Although the authors make a good point that the novel variant in *SPDL1* may not involve in the telomerase pathway, by showing that carriers of *MUC5B* or *SPDL1* did not exhibit statistically significant differences in their telomere lengths compared to the remainder of participants. I still feel that stronger, more direct evidence are needed for explaining the possible new etiology for *SPDL1*

As per response to reviewer 1, we have included gene expression data demonstrating *SPDL1* expression is significantly increased in tissue derived from IPF carriers compared to COPD- and control-derived tissue. We thank both reviewers for their suggestions as these data further strengthen the biological role of *SPDL1* in the pathophysiology of the disease.

Our inclusion of TelSeq data was not to definitively extrapolate the pathophysiology of the *SPDL1* variant, but instead to demonstrate that, similar to *MUC5B* risk allele, the current data suggest it likely acts through a different mechanism that motivates further study. In the first version of the manuscript, we included potential hypotheses for the role of *SPDL1* in the disease aetiology (lines 179-197). We now explicitly mention that these hypotheses need to be followed up with careful functional work (Line 205). The journey from discovering an unequivocal genetic risk factor to comprehensively understanding the underlying pathobiology can be a long one – *MUC5B* being a relevant example of how long this journey can be. Thus, our primary motivation for socialising our *SPDL1* discovery with the community is to ensure that in addition to our own continued interest we crowdsource from the broader academic community—potentially including international researchers who are focused on *SPDL1* biology—to efficiently get us closer to a complete understanding of the disease biology.

References

1. Nalysnyk, L., Cid-Ruzafa, J., Rotella, P. & Esser, D. Incidence and prevalence of idiopathic pulmonary fibrosis: review of the literature. *European Respir Rev* 21, 355–361 (2012).
2. Povysil, G. et al. Rare-variant collapsing analyses for complex traits: guidelines and applications. *Nat Rev Genet* 20, 747–759 (2019).
3. Petrovski, S. et al. An Exome Sequencing Study to Assess the Role of Rare Genetic Variation in Pulmonary Fibrosis. *Am J Resp Crit Care* 196, 82–93 (2017).

REVIEWERS' COMMENTS:

Reviewer #1 (Remarks to the Author):

This paper focus on the genetic pathogenic-associated factors of idiopathic pulmonary fibrosis and identify a novel missense variant in SPDL1 by the application of exome-wide association study (exWAS) and gene-level collapsing analyses. The authors answered my questions and I do not have more doubt.

Reviewer #2 (Remarks to the Author):

The authors have addressed all of my previous comments, although what I meant in my 3rd comment about more direct evidence are, but not limited to 1) gene expression changes among cases and controls; 2) experimental work of that novel variant in animal models; 3) additional in silico analysis on the functional and structural impacts of that variant. The authors have been able to address it to some extent by looking into the publicly available gene expression data.

Reviewer #3 (Remarks to the Author):

Ryan Dhindsa and co-authors present a thorough response to the points raised by Reviewer #1 during the first round of peer review. The consistency of the association between SPDL1 p.Arg20Gln and IPF seen in three independent collections mostly argue against population stratification being a confounder of the association observed. There are several minor points in this revision, which, if they are addressed, will further enhance the accessibility of the manuscript to readers, as well as its enjoyability:

1. I humbly acknowledge the authors' careful attention to detail for the study design that included harmonizing the case-control composition in this study for sex, sequencing coverage, and population stratification. Nonetheless, variant SPDL1 p.Arg20Gln is prone to cryptic population stratification if one examines the frequencies of the minor allele in gnomad. It would be straightforward to fit a Firth penalized logistic regression model (for example, the authors can consider Farhan SM et al., Nature Neuroscience 2019; 22:1966–1974) to put any lingering, residual doubts to rest.
2. Will it be possible for the authors to provide the risk allele frequency of SPDL1 p.Arg20Gln in cases and controls from the FinnGen replication collection? This is because this particular SPDL1 p.Arg20Gln variant has a markedly higher frequency amongst participants of Finnish ancestry compared to participants from elsewhere around the world (https://gnomad.broadinstitute.org/variant/5-169015479-G-A?dataset=gnomad_r2_1). Interested readers would be very keen to know the distribution of the risk allele between cases and controls (and again, they might entertain the very minor possibility of population stratification if not already addressed directly).
3. The authors could consider explaining why Fingerlin TE et al., Nature Genetics 2013 failed to discover this association despite studying >1600 cases and >4600 controls in a discovery GWAS. Was it because of the minor allele frequency cut off used by Fingerlin et al.,? Or the Haplotype Reference Consortium panel that was used by the more recent Allen et al., 2019 (Reference #4 cited by the authors) that helped pick up this low frequency SPDL1 Arg20Gln variant?
4. Most humbly, i found that this sentence could be misleading: "Despite its relatively strong effect size, the SPDL1 locus has not been previously reported in IPF through prior GWAS with larger sample sizes (Table S4)". Might i suggest recasting it to read: "We reassessed the association between SPDL1 p.Arg20Gln in a recently conducted GWAS (Allen et al., 2019) that meta-analyzed 2668 patients with IPF and 8591 unaffected individuals from Chicago, Colorado, and the UK". We found....."

Also, if the sample sizes from Allen et al., for UK, Colorado, and Chicago could be added to Table

S4, this would help the reader understand why the results from UK and Chicago were non-significant, whereas the results from Colorado were genome-wide significant.

5. Do please consider explaining R-sq in Table S4. Was it because Allen et al, used the Haplotype Reference consortium reference panel to impute the low frequency SPDL1 p.Arg20Gln variant, and although the imputation went well (R-sq >0.8), information capture was imperfect? Would this have caused a loss of power, and thus this variant falling short of the genome-wide significant threshold of $P < 5 \times 10^{-8}$ in Allen et al.,?

Reviewer #3 (Remarks to the Author):

Ryan Dhindsa and co-authors present a thorough response to the points raised by Reviewer #1 during the first round of peer review. The consistency of the association between SPDL1 p.Arg20Gln and IPF seen in three independent collections mostly argue against population stratification being a confounder of the association observed. There are several minor points in this revision, which, if they are addressed, will further enhance the accessibility of the manuscript to readers, as well as its enjoyability:

1. I humbly acknowledge the authors' careful attention to detail for the study design that included harmonizing the case-control composition in this study for sex, sequencing coverage, and population stratification. Nonetheless, variant SPDL1 p.Arg20Gln is prone to cryptic population stratification if one examines the frequencies of the minor allele in gnomad. It would be straightforward to fit a Firth penalized logistic regression model (for example, the authors can consider Farhan SM et al., Nature Neuroscience 2019; 22:1966–1974) to put any lingering, residual doubts to rest.

Results from a firth logistic regression correcting for sex, age and the top PCs has now been performed for the SPDL1 allele. In comparison to the exact test $p=2.4 \times 10^{-7}$, a comparable p-value of $p=7.2 \times 10^{-6}$ was achieved. See main text on lines 310 – 313. The p-value among the FinnGen data is already based on a logistic regression correcting for PCs and achieved an independent p-value of $p=1.0 \times 10^{-15}$.

2. Will it be possible for the authors to provide the risk allele frequency of SPDL1 p.Arg20Gln in cases and controls from the FinnGen replication collection? This is because this particular SPDL1 p.Arg20Gln variant has a markedly higher frequency amongst participants of Finnish ancestry compared to participants from elsewhere around the world (https://gnomad.broadinstitute.org/variant/5-169015479-G-A?dataset=gnomad_r2_1). Interested readers would be very keen to know the distribution of the risk allele between cases and controls (and again, they might entertain the very minor possibility of population stratification if not already addressed directly).

The allele frequency of the SPDL1 p.Arg20Gln variant in FinnGen is 0.069 in cases versus 0.030 in controls. This is included in the main text on line 79.

3. The authors could consider explaining why Fingerlin TE et al., Nature Genetics 2013 failed to discover this association despite studying >1600 cases and >4600 controls in a discovery GWAS. Was it because of the minor allele frequency cut off used by Fingerlin et al.,? Or the Haplotype Reference Consortium panel that was used by the more recent Allen et al., 2019 (Reference #4 cited by the authors) that helped pick up this low frequency SPDL1 Arg20Gln variant?

Fingerlin et al. restricted their analyses to study variants with a MAF greater than 0.05. The frequency of this SPDL1 variant in European controls is 0.0078.

4. Most humbly, i found that this sentence could be misleading: "Despite its relatively strong effect size, the SPDL1 locus has not been previously reported in IPF through prior GWAS with larger sample sizes (Table S4)". Might i suggest recasting it to read: "We reassessed the association between SPDL1 p.Arg20Gln in a recently conducted GWAS (Allen et al., 2019) that meta-analyzed 2668 patients with IPF and 8591 unaffected individuals from Chicago, Colorado, and the UK". We found....."

Also, if the sample sizes from Allen et al., for UK, Colorado, and Chicago could be added to Table S4, this would help the reader understand why the results from UK and Chicago were non-significant, whereas the results from Colorado were genome-wide significant.

This variant was not formally assessed in Allen et al., as that analysis only considered variants achieving a p-value < 0.05 across all three component studies. We have now added the sample sizes for the Chicago, UK, and Colorado cohorts to Table S3, highlighting that as the reviewer suggests, the Colorado study was more powered to detect the SPDL1 variant due to larger sample size:

Chicago: 541 cases, 542 controls.

Colorado: 1,515 cases, 4,683 controls.

UK: 612 cases, 3,366 controls

5. Do please consider explaining R-sq in Table S4. Was it because Allen et al, used the Haplotype Reference consortium reference panel to impute the low frequency SPDL1 p.Arg20Gln variant, and although the imputation went well (R-sq >0.8), information capture was imperfect? Would this have caused a loss of power, and thus this variant falling short of the genome-wide significant threshold of $P < 5 \times 10^{-8}$ in Allen et al.,?

This variant was adequately imputed in the Allen et al. study. As mentioned in previous response, Allen et al. only meta-analyzed variants achieving a p-value < 0.05 across all three component cohorts. We include this note in the Table S3 legend for readers.